# Bridging the Gap: How Gender Influences Spanish Politicians' Activity on Twitter

Frederic Guerrero-Solé  and Cristina Perales-García *

Department of Communication, University of Pompeu Fabra, 08002 Barcelona, Spain; frederic.guerrero@upf.edu
* Correspondence: cristina.perales@upf.edu

**Abstract:** Women have historically been underrepresented in politics. However, in the last few decades, more and more women have been elected to both upper and lower houses, particularly in Spain. Social media has become one >of the main channels for women to gain visibility, but the issue of unequal distribution of power and influence between men and women remains. This paper sheds light on gender differences among politicians on Twitter by analyzing the social media activity and influence of 277 of the 350 Members of the Spanish Congress of Deputies from March to June 2020. Our research shows there are still major gender differences regarding audience figures and amplification and that both male and female politicians still largely retweet more men than women. In addition, we found significant differences between parties and across the political spectrum, although these are less prominent (albeit not neutralized) in parties with a female leader. This is in keeping with studies that have found broad similarities between male and female politicians' communicative practices, but a persistently large gap to be bridged in terms of their online influence. Female leaders are proposed as a means to bridge this gap.

**Keywords:** gender; Twitter; politics; Spain; female politicians



## 1. Introduction

### 1.1. Gender Differences in Political Power and Influence, and Underrepresentation and Empowerment in Politics

Politics, like many other areas of human activity, has traditionally been male-dominated territory. Female representation in democratic parties, congresses, senates and powerful political offices has only increased in the last few decades (Elder 2020; Bridgewater and Nagel 2020). However, research has shown that the situation is still far from balanced. Although women have populated parties, local governments and parliaments, various studies have revealed that sexism in the culture of political parties tends to favor male candidates on the ballot, to systematically disempower women (Verge and Troupel 2011; Verge and de la Fuente 2014) and to hamper women's access to powerful political offices (Lovenduski 2005; Verge 2010). According to Verge and Wiesehomeier (2019), such discrimination runs across all parties, and parity is still a long way off, even in the most representative democracies.

Spain is a particular case of a country in which women have been historically underrepresented in politics (Fernández and Eugenia 2008). In 1977, two years after dictator Francisco Franco's death, the constituent legislature had 21 female Members (5.8% of the total). This number barely increased in the first legislature in 1979, in which a mere 24 out of 350 Members were women, none of whom held any significant office in government. By the terms of 1989 and 1993, the proportion of female Members in the Spanish Congress had reached a meagre 10%. However, in 1996, almost a hundred women (23.9%) were elected in the first People's Party government. In 2007 a new Equality Law came into force, requiring political parties to ensure minimum gender representation of 40% in candidates running for office (Verge 2010). This helped to balance the male-dominated political culture (see Table 1) visible in Spanish politics since the transition to democracy (Valiente 2008;

Verge 2012). Nevertheless, Verge and Wiesehomeier (2019) argue that discrimination did not suddenly disappear with the 2007 quota. Although quotas tend to balance gender representation, other barriers to women in the political sphere, such as having to conform to male norms (Verge and de la Fuente 2014, p. 71), cause many women to relinquish certain offices (Verge 2015). Notwithstanding the ongoing inequality in Spanish politics, the number of elected women has increased in the last decade, and Spain now ranks sixteenth in the world in terms of women's representation in parliament (Verge and Wiesehomeier 2019).

**Table 1.** Percentage of women in Spanish Congress in 2019 per party *.

| Party | % Women |
| --- | --- |
| Vox | 26.9 |
| PP | 43.2 |
| PSOE | 48.3 |
| Cs | 50.0 |
| JxCat | 50.0 |
| UP | 51.4 |
| ERC | 53.8 |

Source: INE (Available online: https://www.ine.es/jaxi/Tabla.htm?path=/t00/mujeres_hombres/tablas_1/l0/&file=p02001.px Accessed on 28 March 2021). * Left and left-of-center parties: UP = Unidas Podemos/United We Can; PSOE = Partido Socialista Obrero Español/Spanish Socialist Workers' Party; ERC = Esquerra Republicana de Catalunya/Republican Left of Catalonia. Right-of-center and liberal parties: JxCAT = Junts per Catalunya/Together for Catalonia; Cs = Ciudadanos /Citizens; PP = Partido Popular/People's Party. Far-right parties: Vox.

The aim of this research is to analyze gender differences in Twitter use among Spanish members of parliament. To do so, we gathered all tweets from 277 of the 350 Members of the Spanish Congress from March to June 2020. We measured four variables related to their overall Twitter use: number of tweets, mean number of followers (audience), number of retweets (amplification), and efficacy. In addition, we measured the number of times that Members were retweeted by fellow party members (internal amplification), which can be linked to the internal communication strategies of the parties analyzed.

*1.2. Communicating for Influence and Visibility*

One way that women can increase their visibility in society is to garner media coverage. Representation in the media allows women to normalize their role in politics while also allowing for an impact on the political agenda (Kreiss 2016) as well as to articulate policy positions (Sobieraj et al. 2020). However, the media have traditionally under- or misrepresented women (Wasburn and Wasburn 2011; Sánchez Calero et al. 2013; Lünenborg and Maier 2015; Larson 2001; Fernández García 2013; Guerrero-Solé 2018; Dunaway et al. 2013). Currently, social networks are at the heart of all political communications strategies (Usher et al. 2018). Politicians the world over utilize social media, not only during electoral campaigns, but also for everyday communications (Graham et al. 2016). Politicians' use of social media is a strategic form of publicity (Kreiss 2016; Cervi and Roca 2017; Casero-Ripollés et al. 2020; Guerrero-Solé and Lluís 2017; Guerrero-Solé and López-González 2019). As a consequence, politicians' influence is no longer estimated exclusively on the basis of their coverage in traditional media, but also on their popularity on social networks, where follower numbers, shares, retweets and likes are the measure of their success. Politicians' activity on social networks is also considered to be a driver for media attention (Rauchfleisch and Metag 2020; Graham et al. 2016). Social media activity is therefore a priority for female politicians, particularly given that research has shown they receive less media attention (Miller and Peake 2013; Baitinger 2015; Tromble and Koole 2020) and more negative coverage (Armstrong and Gao 2011; Ross et al. 2013; Larson 2001) than their male counterparts. McGregor and Mourão (2016) hold that women are more central to the conversation about them and about their opponents than men; this indicates that their connections in social networks are stronger. Various studies suggest that women having

more visibility on social networks and communicating directly with citizens (Loiseau and Nowacka 2015; Vergeer 2015) can help to redress this.

### 1.3. The Role of Gender and Party on Twitter

Twitter has become one of the main tools that politicians use to complement their traditional communication strategies (Jungherr and Schoen 2013; Vergeer et al. 2013; Jungherr 2014). But what role does gender play in female politicians' activity and influence on Twitter? Gender research into social media focuses mainly on two areas: harassment of women on social networks and the differing topics that men and women talk about. With regard to the former, the results to date are inconclusive and culture-specific. Some researchers have concluded that female politicians face more negativity on social media than traditional media (Conroy et al. 2015) and are more likely to be the target of hate speech (Wilhelm and Joeckel 2018) or uncivil tweets questioning their positions as politicians (Southern and Harmer 2019). On the other hand, Tromble and Koole (2020) found that in the UK, US and the Netherlands, gendered insults are infrequent. In relation to the second area, past research has found that female politicians tend to talk more about issues that predominantly affect women (Pearson and Dancey 2011). Moreover, although there are only minor gender differences in communication styles in some cases (Hrbková and Macková 2020), in general gender and party have an effect on what women tweet about (Hemphill et al. 2020; Johnstonbaugh 2020; Evans and Clark 2016).

In addition to harassment and styles of communication, research has also been carried out on the following: the gendered distribution of relational power in network discussions (McGregor and Mourão 2016); different patterns of liking practices; support of issues and civic engagement (Brandtzaeg 2017); self-presentation on social networks (Cook 2016); gender stereotypes of politicians online (Beltran et al. 2020; Wagner et al. 2017).

However, few studies have focused on gender and party differences in politicians' number of tweets, size of audience, amplification and efficacy. We believe that this analysis can offer significant insight into the extent to which Twitter evens out any such hypothetical differences between men and women. Consequently, our first research question is as follows:

RQ1: Are there gender differences among Spanish Members with regard to number of tweets, audience, amplification and efficacy on Twitter?

As we have already mentioned, gender is not the only variable that might explain differences between politicians. Party membership can also be a predictor of politicians' activity and influence in online environments (Johnstonbaugh 2020).
Therefore, the second research question is

RQ2: Are there party differences among Spanish Members with regard to number of tweets, audience, amplification and efficacy on Twitter? Are there differences between left- and right-wing parties?

In Spain, left-wing parties have strived to achieve gender equality (Uribe Otalora 2013). Therefore, male–female internal amplification can be a measure of how much attention fellow Members pay their female and male colleagues and whether they are equally likely to retweet them. Thus, the third research question is

RQ3: Is the amplification rate among female and male Spanish Members balanced?

## 2. Sample and Method

To answer the aforementioned research questions, we gathered all tweets, replies and retweets that Spanish Members posted on Twitter from 14 March to 19 June 2020. This period coincides with the COVID-19 state of alarm in Spain. The sample included 277 out of the 350 Members of the fourteenth legislature, of whom 44% were women and 56% men, from the parties shown in Table 2. They collectively posted 249,874 tweets and retweets in the three months, with an individual minimum of 2 and maximum of 7767 posts.

## 2.1. Independent Variables

We coded for the following independent variables:

**Gender:** gender of the Member (male = 151, female = 126).
**Political party:** political party of the Member (see Table 1).
**Political leaning:** political leaning (left or right) of the Member's party (left = 135, right = 121, independent = 21).

## 2.2. Dependent Variables

The dependent variables were defined as follows:

**Amount:** number of tweets and replies that each Member posted in the period analyzed (min = 0; max = 3045; mean = 259; SD = 341).
**Amplification:** number of times each Member was retweeted during the period (min = 0; max = 1,427,478; mean = 38,412; SD = 128,870).
**Audience:** mean number of each Member's followers during the period (min = 137; max = 1,351,574; mean = 38,270; SD = 136,211).
**Efficacy:** defined as amplification divided by amount and audience (min = 0; max = 209.55; mean = 6.86; SD = 14.06).
**Internal amplification**: proportion of retweets by fellow Members from the same party.

**Table 2.** Breakdown of Spanish Members by party.

| Party | N | Female | Male |
|---|---|---|---|
| UP | 33 | 17 | 16 |
| ERC | 13 | 7 | 6 |
| PSOE | 102 | 49 | 53 |
| JxCat | 8 | 4 | 4 |
| Cs | 9 | 5 | 4 |
| PP | 72 | 31 | 41 |
| VOX | 40 | 13 | 27 |
| Other | 18 | 3 | 15 |
| Total | 295 | 129 | 166 |

## 3. Results

To answer research question one, we first calculated the mean values of the dependent variables: amount, amplification, audience and efficacy. Table 3 shows the mean values by gender of these variables. We performed ANOVA tests to evaluate the statistical significance of the differences between genders.

**Table 3.** Mean values of amount of tweets, number of followers and efficacy of Spanish Members on Twitter by gender.

| | Mean (SD) | | |
|---|---|---|---|
| | **Male** | **Female** | **Sign.** |
| **Amount** | 269 (342) | 247 (341) | 0.608 |
| Retweets published | 641 (952) | 646 (875) | 0.962 |
| Posts published | 909 (1133) | 893 (1110) | 0.907 |
| **Amplification** | 46,349 (149,153) | 28,901 (99,026) | 0.263 |
| **Audience** | 52,397 (174,181) | 21,339 (63,475) | 0.059 |
| **Efficacy** | 7.21 (17.87) | 6.44 (14.06) | 0.654 |

To answer research question two, we calculated the mean values of the dependent variables for each of the seven main parties in the Spanish Congress of Deputies. First, we analyzed the differences in tweet amount, amplification, efficacy and audience (Table 4).

As above, we performed ANOVA tests for statistical differences. The results are also shown in Figure 1.

**Table 4.** Mean amount, amplification, efficacy and audience of Spanish Members by gender and party.

| | Amount | | Amplification | |
|---|---|---|---|---|
| **Party** | **Female** | **Male** | **Female** | **Male** |
| UP | 208 (144) | 231 (228) | 21,877 (47,071) | 70,310 (160,379) |
| ERC | 222 (314) | 226 (139) | 3040 (3070) | 59,413 (138,982) |
| PSOE | 190 (197) | 220 (267) | 6190 (16,343) | 17,516 (46,442) |
| JxCat | 855 (1462) | 287 (198) | 54,550 (72,652) | 62,832 (117,001) |
| Cs | 452 (340) | 593 (549) | 48,826 (69,123) | 76,490 (45,708) |
| PP | 226 (238) | 169 (164) | 43,920 (123,292) | 20,980 (89,027) |
| VOX | 315 (317) | 497 (572) | 86,247 (223,851) | 117,460 (287,975) |
| | Efficacy | | Audience | |
| UP | 5.04 (4.51) | 6.76 (7.47) | 23,543 (31,506) | 132,268 (294,564) |
| ERC | 4.91 (3.02) | 1.89 (0.83) * | 5797 (6059) | 133,505 (304,188) |
| PSOE | 4.66 (6.62) | 4.88 (5.19) | 11,128 (18,067) | 54,672 (203,697) |
| JxCat | 5.79 (5.55) | 4.46 (2.61) | 52,955 (60,473) | 40,346 (69,686) |
| Cs | 5.23 (5.30) | 8.92 (5.71) | 140,063 (279,352) | 27,929 (24,495) |
| PP | 8.77 (9.19) | 5.16 (4.60) * | 16,784 (41,930) | 16,166 (66,196) |
| VOX | 11.26 (8.42) | 16.67 (40.39) | 20,781 (34,141) | 43,006 (95,364) |

* $p < 0.05$.

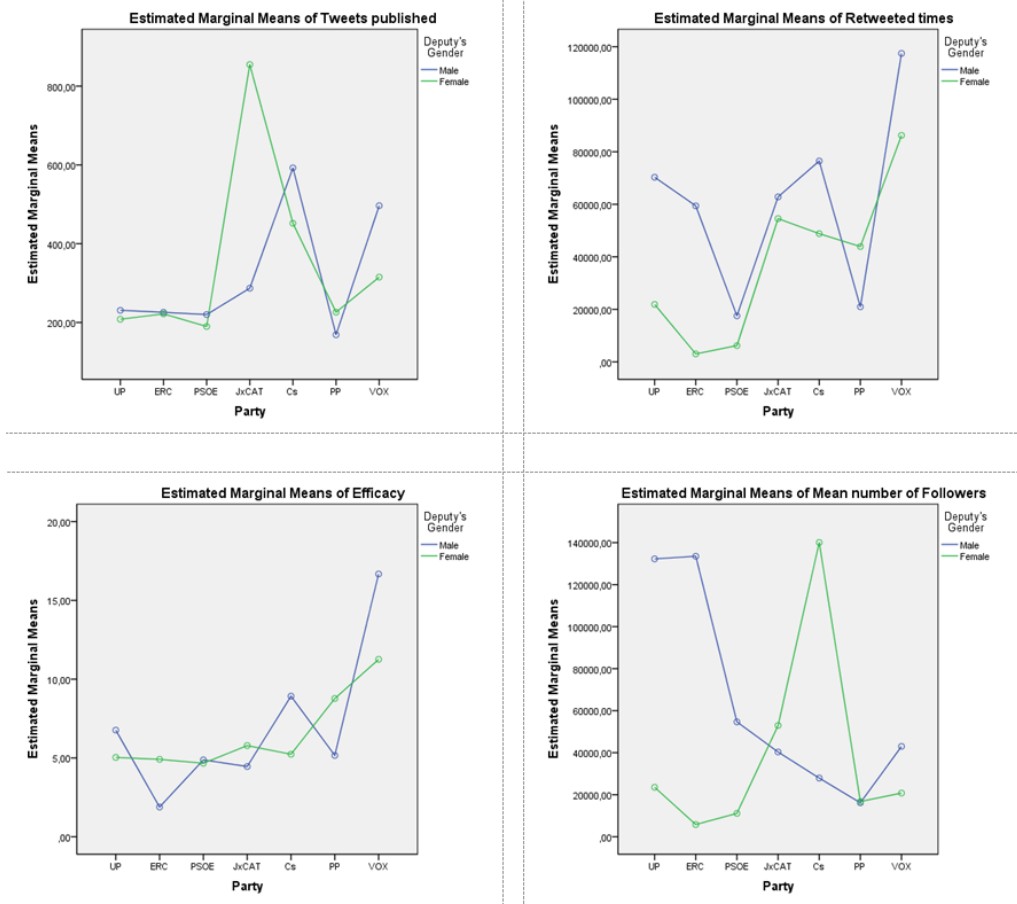

**Figure 1.** Mean amount, amplification, efficacy and audience of Spanish Members by gender and party.

The second part of RQ2 aimed to ascertain differences between left- and right-wing parties in Spain. For this purpose, we labelled UP and PSOE Members as 'left-wing' and Cs, PP and Vox Members as 'right-wing'. We calculated the mean scores of the dependent variables: amount, amplification, audience and efficacy (Table 5).

**Table 5.** Mean amount, amplification, audience and efficacy by political spectrum (left/right).

| Party | Mean (SD) | | |
| --- | --- | --- | --- |
| | **Left (N = 135)** | **Right (N = 121)** | ***p*-Value** |
| **Amount** | 643 (617) | 1116 (1358) | 0.000 |
| **Amplification** | 20,211 (66,718) | 58,384 (176,207) | 0.020 |
| **Audience** | 44,144 (165,751) | 28,318 (85,225) | 0.346 |
| **Efficacy** | 5.05 (5.96) | 9.39 (20.11) | 0.018 |

Whereas the first two questions were related to the general Twitter activity of the Spanish Members, and amplification was defined as how many times they were retweeted overall, RQ3 explored how often Members retweeted posts published by fellow party Members (internal amplification). Table 6 below shows the gender breakdown of intra-party retweets.

**Table 6.** Mean internal amplification of women and men, standard deviation, and significance by gender of the retweeter.

| Gender | IA of Women (SD) | IA of Men (SD) | Signif. |
| --- | --- | --- | --- |
| Male (N = 151) | 55.16 (116.36) | 123.35 (202.94) | 0.000 |
| Female (N = 126) | 62.51 (104.15) | 97.73 (125.83) | 0.000 |
| Total (N = 277) | 58.50 (110.83) | 111.70 (172.38) | 0.000 |

As Table 7 shows, male politicians retweet other male politicians twice as much as they retweet female politicians. Furthermore, female politicians also retweet male politicians more frequently, although the difference is slightly smaller.

**Table 7.** Internal amplification (IA) among Spanish Members by party.

| Party | IA to Women | IA to Men | Sign. | Norm. W | Norm. M |
| --- | --- | --- | --- | --- | --- |
| UP (N = 33) | 17.21 (15.68) | 33.67 (34.92) | 0.001 | 0.51 | 1 |
| ERC (N = 13) | 65.00 (60.50) | 117.08 (111.01) | 0.016 | 0.56 | 1 |
| PSOE (N = 102) | 35.95 (50.69) | 67.43 (73.02) | 0.000 | 0.53 | 1 |
| **JxCat (N = 8)** | **37.38 (28.85)** | **53.63 (41.27)** | **0.221** | **0.70** | **1** |
| **Cs (N = 9)** | **81.89 (87.93)** | **91.00 (105.76)** | **0.749** | **0.90** | **1** |
| PP (N = 72) | 98.11 (172.66) | 149.47 (200.14) | 0.000 | 0.66 | 1 |
| Vox (N = 40) | 75.63 (129.39) | 235.48 (294.53) | 0.000 | 0.32 | 1 |
| Total (N = 277) | 58.83 (114.39) | 113.24 (177.28) | 0.000 | 0.52 | 1 |

We performed a detailed analysis of the internal amplification strategies of men and women by party and gender (Table 8) and found that, in all cases, both men and women retweeted more tweets from men than from women. We performed a *t*-Test for paired samples and found that in the ruling party PSOE, the right-wing PP, and the far-right party Vox, the gender differences were highly significant.

**Table 8.** Internal amplification (IA) of women and men by party and gender.

| | Women | | | Men | | |
| --- | --- | --- | --- | --- | --- | --- |
| **Party** | **IA of Women** | **IA of Men** | **Signif.** | **IA of Women** | **IA of Men** | **Signif.** |
| UP | 12.82 (11.46) | 24.35 (26.59) | 0.053 | 21.88 (18.41) | 43.56 (40.55) | 0.011 |
| ERC | 44.57 (51.30) | 101.00 (136.51) | 0.142 | 88.83 (66.04) | 135.83 (80.20) | 0.026 |
| PSOE | 51.42 (62.11) | 78.47 (87.73) | 0.001 | 21.64 (31.55) | 57.23 (55.04) | 0.000 |
| **JxCAT** | **56.50 (28.87)** | **85.25 (34.24)** | **0.288** | **18.25 (11.53)** | **22.00 (11.63)** | **0.704** |
| **Cs** | **49.00 (66.97)** | **58.20 (66.21)** | **0.261** | **123.00 (102.87)** | **132.00 (141.24)** | **0.901** |
| PP | 122.97 (177.43) | 157.1935 (190.10) | 0.102 | 79.32 (168.71) | 143.63 (209.58) | 0.000 |
| Vox | 41.77 (38.38) | 141.77 (103.51) | 0.001 | 91.93 (153.57) | 280.59 (344.49) | 0.000 |

## 4. Discussion

Research has shown that women have historically been discriminated against in politics. Unequal distribution of political positions and responsibilities coupled with women's underrepresentation in parliaments have driven the need for gender quotas (Verge 2010; Verge and de la Fuente 2014). This has resulted in significantly more women in parties and governments than in the past. However, parity is still a long way off, particularly due to the underlying androcentric political culture in some countries. Spain has been no exception when it comes to a gender imbalance in politics, and women have achieved increased visibility and power only in the last decade. The media have often spearheaded this shift, and today social media is one way that enables women to increase their presence, power and visibility. However, the issue of equality remains.

Our research analyzed the extent to which male and female Members of the Spanish Congress are equally influential in terms of content amount, amplification, audience and efficacy on Twitter, one of the most widely used social networks for political communications in Spain. The results show that there are few overall gender differences when it comes to number of tweets. We found that male and female Members are equally active on Twitter, which is in tune with the reported increase in women's visibility on social networks (Loiseau and Nowacka 2015; Vergeer 2015). Our results also echo previous studies that have found minor differences in candidate online campaigning coverage (Tromble and Koole 2020) and reveal Spanish female politicians' effort to be as active and influential on social networks as men. However, we found major disparities in the amplification of tweets (men are retweeted twice as many times as women) and audience (men have more than double the audiences of women). Nevertheless, most of these differences were not statistically significant due to the skewed distribution of variables (see Table A1 in Appendix A for the scores of variables for each Member).

When we broke down the analysis by party, the only considerable gender difference in amount of tweets was in the female-led Catalan party JxCat (women tweeted three times more than men) and the populist far-right party Vox (men tweeted twice as much as women). With regard to the other variables analyzed, we found that gender differences in amplification were notable, in particular in UP and ERC. In all parties except the right-wing PP, women were less amplified on the network than men. These results are in tune with previous research on the interaction of party and gender stereotypes on politicians' effectiveness when they use Twitter (Holman et al. 2011). There were also stark differences in audience in the female-led party Cs. Finally, we found statistically significant differences between men and women in efficacy in ERC and UP. While the UP male Members' efficacy was significantly greater than the women's, in ERC, women had almost three times the efficacy of their male counterparts. The case of UP is significant because it defines itself as a feminist party and has clearly feminist policies. However, as the overall results show, women remain a minority in the male-dominated political sphere.

Statistically significant differences emerged when we grouped parties by ideological leaning. The right-wing parties Cs, PP and Vox were far more active than left-wing parties UP and PSOE. The same was true of amplification and efficacy, although the differences were lesser ($p < 0.05$). Amplification in right-wing parties was three times greater than in left-wing parties; efficacy was twice as high, and mean audience was almost half. In short, the right-wing parties, currently in the opposition, were far more active, had a greater

impact on the network, and were much more efficient than the ruling left-wing parties. These results suggest that party and ideological leaning are better predictors of differences than gender in content amount, amplification and efficacy.

However, the most relevant and interesting results of this research are for internal amplification according to political party. We found that in all seven parties analyzed, internal amplification of men was substantially larger (broadly double) than of their female counterparts. Moreover, in five parties this difference was statistically significant. Earlier research found a sexist and discriminatory culture in most parties that favors male candidates on ballots, systematically disempowers women (Verge and Troupel 2011; Verge and de la Fuente 2014) and hampers women's access to relevant political positions (Lovenduski 2005; Verge 2010). It is interesting to note that the two parties in which gender differences in internal amplification were not statistically significant (JxCat and Cs) were both led by a woman. It is therefore possible to conclude that having a female leader, i.e., allowing women to access relevant political positions, may balance out differences in internal amplification.

The results are similar when we look at internal amplification by gender. Women internally amplify more men than women, although the differences are only statistically significant in the ruling party PSOE and the far-right party Vox. Men also retweet more male than female fellow party members. Again, all of the differences observed are significant except for JxCat and Cs, the two parties in the Spanish Congress of Deputies with female leaders. We can therefore conclude that women are broadly discriminated against in the internal communications strategies of political parties in Spain on Twitter, especially in the case of women who are discriminated against by male party colleagues. This discrimination is not related to the party's position on the political spectrum and is only neutralized in female-led parties. These results confirm previous findings that show that Twitter is far from being a public sphere in which gender inequalities are eliminated (Hu and Kearney 2020).

Finally, it is worth mentioning that this research was performed with a sample of tweets collected during the first COVID-19 state of alarm in Spain. There is evidence that contexts with heightened states of national security threat—and the COVID-19 outbreak may be considered such a case—can activate preferences for male politicians (Holman et al. 2011). Consequently, new research is needed in the future to support and generalize the conclusions of our work.

**Author Contributions:** Conceptualization: F.G.-S. and C.P.-G.; Data curation: F.G.-S.; Investigation: F.G.-S.; Methodology: F.G.-S. and C.P.-G.; Supervision: C.P.-G.; Validation: F.G.-S. and C.P.-G.; Writing—original draft: F.G.-S.; Writing—review & editing: F.G.-S. and C.P.-G. All authors have read and agreed to the published version of the manuscript.

**Funding:** This research was funded by the MCIU/AEI/FEDER, UE under Grant PGC2018-097352-A-I00.

**Institutional Review Board Statement:** Not applicable.

**Informed Consent Statement:** Not applicable.

**Data Availability Statement:** Not applicable.

**Conflicts of Interest:** The authors declare no conflict of interest.

## Appendix A

**Table A1.** Twitter data of the 277 Spanish Members in the sample.

| Gender | Party | Twitter Handle | Followers | Activity | Retweets | RT Times |
|--------|-------|----------------|-----------|----------|----------|----------|
| F | Cs | inesarrimadas | 638,783 | 476 | 249 | 168,040 |
| F | Cs | mariadelamiel | 45,465 | 1071 | 163 | 49,758 |
| F | Cs | martamartirio | 10,242 | 2672 | 1947 | 16,115 |
| F | Cs | mcmartinez_cs | 1094 | 521 | 288 | 3259 |
| F | Cs | saragimnez | 4733 | 284 | 118 | 6958 |
| M | Cs | baledmundo | 28,279 | 802 | 623 | 74,017 |
| M | Cs | guillermodiazcs | 15,015 | 2790 | 1391 | 65,040 |
| M | Cs | marcosdequinto | 62,087 | 515 | 71 | 138,377 |
| M | Cs | paucambronerocs | 6333 | 2281 | 1933 | 28,526 |
| F | ERC | bassamontse | 11,445 | 168 | 85 | 8286 |
| F | ERC | caroltelechea | 2606 | 59 | 32 | 531 |
| F | ERC | inesgranollers | 987 | 1498 | 1202 | 2065 |
| F | ERC | martarosiq | 16,775 | 311 | 184 | 4390 |
| F | ERC | normapujol | 2373 | 433 | 365 | 400 |
| F | ERC | pilarvallugera | 1044 | 369 | 322 | 265 |
| F | ERC | _maria_dantas_ | 5346 | 4710 | 3806 | 5344 |
| M | ERC | capdevilajoan | 8306 | 1528 | 1398 | 1668 |
| M | ERC | gabrielrufian | 754,249 | 1334 | 1145 | 343,089 |
| M | ERC | joanmargall | 4346 | 1420 | 1151 | 3225 |
| M | ERC | jsalvadorduch | 7109 | 432 | 304 | 1538 |
| M | ERC | nuet | 23,466 | 2612 | 2124 | 5654 |
| M | ERC | xavieritja | 3555 | 914 | 763 | 1302 |
| F | JxCAT | conceptermens | 939 | 370 | 314 | 694 |
| F | JxCAT | lauraborras | 114,448 | 7033 | 3988 | 154,404 |
| F | JxCAT | marionaid | 1105 | 742 | 648 | 684 |
| F | JxCAT | miriamnoguerasm | 95,328 | 1587 | 1363 | 62,417 |
| M | JxCAT | ferran_bel | 7698 | 875 | 515 | 6943 |
| M | JxCAT | genisboadella | 2498 | 416 | 326 | 1859 |
| M | JxCAT | jacs_jaumeacs | 144,823 | 823 | 293 | 238,307 |
| M | JxCAT | sergimiquel | 6365 | 293 | 125 | 4221 |
| F | PP | abeltran_ana | 8121 | 374 | 204 | 29,416 |
| F | PP | aliciagarcia_av | 4603 | 1141 | 716 | 10,699 |
| F | PP | anadebande | 12,228 | 4131 | 3196 | 59,218 |
| F | PP | anapastorjulian | 106,286 | 730 | 417 | 133,732 |
| F | PP | anazurita7 | 3449 | 645 | 519 | 3584 |
| F | PP | auxipd | 1462 | 180 | 180 | 0 |
| F | PP | bealinuesa | 1553 | 336 | 160 | 454 |
| F | PP | bea_fanjul | 59,436 | 735 | 306 | 580,147 |
| F | PP | belenhoyo | 11,017 | 593 | 419 | 4462 |
| F | PP | borrego_corte | 1679 | 923 | 914 | 143 |
| F | PP | carmenriolobos | 6262 | 3721 | 3029 | 3534 |
| F | PP | carolinaespanar | 3192 | 449 | 440 | 96 |
| F | PP | cayetanaat | 212,899 | 319 | 163 | 389,637 |
| F | PP | cnlacoba | 1901 | 223 | 140 | 547 |
| F | PP | cucagamarra | 10,328 | 595 | 362 | 14,384 |
| F | PP | edurneuriarte | 21,967 | 213 | 24 | 62,997 |
| F | PP | llanosdeluna | 673 | 202 | 182 | 494 |
| F | PP | margaprohens | 5743 | 3211 | 2643 | 10,416 |
| F | PP | mariaramallov | 433 | 51 | 30 | 17 |
| F | PP | martaglezvzqz | 7841 | 32 | 21 | 374 |
| F | PP | mdelaoredondo | 284 | 160 | 154 | 15 |
| F | PP | milamarcos | 2033 | 1304 | 1051 | 951 |
| F | PP | moromjesus | 3796 | 2463 | 2249 | 2118 |
| F | PP | palomagazquez | 2098 | 4816 | 4768 | 3307 |
| F | PP | pilarmarcosd | 4660 | 2505 | 2076 | 8433 |

**Table A1.** *Cont.*

| Gender | Party | Twitter Handle | Followers | Activity | Retweets | RT Times |
|--------|-------|----------------|-----------|----------|----------|----------|
| F | PP | rosaromerocr | 9571 | 988 | 278 | 13,764 |
| F | PP | solcruzguzman | 2145 | 683 | 475 | 1310 |
| F | PP | tejerinapp | 2180 | 32 | 32 | 0 |
| F | PP | teresajbecerril | 6645 | 324 | 119 | 20,190 |
| F | PP | tristanamg | 2593 | 414 | 407 | 109 |
| F | PP | valentinam | 3227 | 317 | 125 | 6975 |
| M | PP | aalmodobar | 4093 | 1393 | 642 | 4131 |
| M | PP | aglezterol | 19,432 | 431 | 160 | 35,821 |
| M | PP | albertocasero | 2147 | 1294 | 1173 | 715 |
| M | PP | andreslorite | 3822 | 832 | 495 | 10,697 |
| M | PP | carlosrojas_ppa | 5440 | 1321 | 1233 | 3064 |
| M | PP | celsodelgadoou | 1240 | 141 | 126 | 28 |
| M | PP | diegogagob | 7556 | 434 | 341 | 3219 |
| M | PP | diegomovellan | 1621 | 696 | 639 | 1481 |
| M | PP | educarazo | 2967 | 628 | 353 | 2445 |
| M | PP | eloysuarezl | 4786 | 451 | 149 | 2669 |
| M | PP | gmariscalanaya | 6065 | 439 | 412 | 1075 |
| M | PP | herrerobono | 4830 | 171 | 36 | 826 |
| M | PP | hispanpablo | 855 | 27 | 14 | 164 |
| M | PP | jacallejascano | 629 | 307 | 136 | 558 |
| M | PP | jaimedeolano | 13,456 | 2682 | 2317 | 30,005 |
| M | PP | jangelvillalon | 2032 | 283 | 207 | 569 |
| M | PP | javierbasco | 332 | 236 | 231 | 0 |
| M | PP | javier_merino | 2463 | 397 | 259 | 638 |
| M | PP | jiechaniz | 3146 | 574 | 118 | 27,000 |
| M | PP | josemiguel_glez | 379 | 68 | 64 | 4 |
| M | PP | jspostigo | 656 | 430 | 403 | 31 |
| M | PP | juan_pedreno | 175 | 42 | 22 | 5 |
| M | PP | luisstamaria | 4174 | 339 | 273 | 509 |
| M | PP | mapaniagua | 4532 | 189 | 83 | 1300 |
| M | PP | mariogarcessan | 4339 | 172 | 50 | 7477 |
| M | PP | mcastellonpp | 1523 | 346 | 232 | 243 |
| M | PP | miqueljerez | 1617 | 656 | 607 | 651 |
| M | PP | montesinospablo | 40,420 | 537 | 357 | 21,305 |
| M | PP | oscarclavell | 2905 | 48 | 32 | 211 |
| M | PP | oscargamazo | 1702 | 1528 | 1174 | 679 |
| M | PP | otazu35 | 696 | 1097 | 937 | 1533 |
| M | PP | pablocasado_ | 423,738 | 760 | 292 | 562,173 |
| M | PP | pedronavarrol | 2299 | 1686 | 1306 | 1492 |
| M | PP | quin1954 | 2382 | 82 | 66 | 212 |
| M | PP | sanchezcesar | 8575 | 155 | 83 | 748 |
| M | PP | sebastianlede15 | 691 | 1146 | 1055 | 119 |
| M | PP | tcabcas | 1080 | 635 | 561 | 689 |
| M | PP | teogarciaegea | 61,518 | 434 | 214 | 123,982 |
| M | PP | vicentebetoret | 5399 | 627 | 478 | 2953 |
| M | PP | vicentetiradopp | 1548 | 6470 | 6360 | 1083 |
| M | PP | vicpiriz1975 | 5527 | 876 | 443 | 7693 |
| F | PSOE | adrilastra | 81,460 | 566 | 465 | 107,381 |
| F | PSOE | afernb | 12,972 | 924 | 344 | 25,213 |
| F | PSOE | anaprietonieto | 8163 | 3753 | 2895 | 8084 |
| F | PSOE | angelesmarra | 959 | 536 | 301 | 251 |
| F | PSOE | ariagonagp | 522 | 22 | 14 | 4 |
| F | PSOE | beamcarrillo | 2611 | 413 | 299 | 821 |
| F | PSOE | beatrizcorredor | 13,269 | 967 | 820 | 1172 |
| F | PSOE | begonasarre | 2332 | 882 | 574 | 755 |
| F | PSOE | belenfcasero | 1775 | 701 | 631 | 1278 |
| F | PSOE | belitagl | 760 | 854 | 796 | 289 |
| F | PSOE | caballerohelena | 577 | 1537 | 1144 | 363 |

**Table A1.** *Cont.*

| Gender | Party | Twitter Handle | Followers | Activity | Retweets | RT Times |
|--------|-------|----------------|-----------|----------|----------|----------|
| F | PSOE | carmenandres_ | 3912 | 323 | 267 | 346 |
| F | PSOE | carmencalvo_ | 66,863 | 325 | 272 | 17,196 |
| F | PSOE | celaaisabel | 35,923 | 144 | 33 | 16,063 |
| F | PSOE | elviraramon | 3471 | 2365 | 2109 | 1612 |
| F | PSOE | estherpadillar | 3271 | 571 | 454 | 796 |
| F | PSOE | estherpcamarero | 4015 | 303 | 170 | 866 |
| F | PSOE | evabravobarco | 731 | 85 | 33 | 406 |
| F | PSOE | evapatriciab | 649 | 92 | 51 | 220 |
| F | PSOE | fuensantalima | 2667 | 1588 | 1093 | 957 |
| F | PSOE | graciacanales3 | 563 | 182 | 114 | 62 |
| F | PSOE | hernanzsofia | 4698 | 397 | 360 | 452 |
| F | PSOE | lauraberja86 | 3240 | 1238 | 1030 | 4283 |
| F | PSOE | lidiaguinart | 4626 | 1037 | 634 | 4492 |
| F | PSOE | luisacarcedo | 10,310 | 87 | 53 | 5028 |
| F | PSOE | luzseijo | 7184 | 357 | 178 | 7110 |
| F | PSOE | maraluisavilch1 | 180 | 294 | 277 | 16 |
| F | PSOE | marina_ortega_ | 1140 | 585 | 366 | 4564 |
| F | PSOE | maritxu30 | 1810 | 121 | 14 | 513 |
| F | PSOE | marotoreyes | 13,933 | 556 | 388 | 7636 |
| F | PSOE | marrodanmaria | 816 | 5 | 4 | 34 |
| F | PSOE | merceperea | 5605 | 2028 | 1498 | 4281 |
| F | PSOE | meritxell_batet | 49,523 | 583 | 243 | 11,290 |
| F | PSOE | mjmonteroc | 41,120 | 11 | 3 | 1311 |
| F | PSOE | montseminguez | 4019 | 609 | 475 | 1025 |
| F | PSOE | msolsj | 2866 | 2007 | 1677 | 2099 |
| F | PSOE | mvalerio_gu | 21,199 | 791 | 764 | 2414 |
| F | PSOE | nvillagrasa | 1284 | 304 | 188 | 346 |
| F | PSOE | olgaalonso62 | 155 | 551 | 398 | 74 |
| F | PSOE | patri_blanquer | 1644 | 333 | 254 | 1161 |
| F | PSOE | pilicancela | 6243 | 1346 | 673 | 8824 |
| F | PSOE | rafi_crespin | 2827 | 141 | 76 | 207 |
| F | PSOE | sandrage76 | 1028 | 2397 | 2183 | 827 |
| F | PSOE | soniafetesoro | 2786 | 53 | 20 | 41 |
| F | PSOE | ssumelzo | 21,687 | 441 | 398 | 1222 |
| F | PSOE | susana_ros | 6386 | 528 | 435 | 1685 |
| F | PSOE | tamarayar | 1846 | 309 | 293 | 91 |
| F | PSOE | teresaribera | 44,329 | 432 | 229 | 11,675 |
| F | PSOE | zaidacantera | 35,318 | 1923 | 1303 | 36,460 |
| M | PSOE | abalosmeco | 70,836 | 403 | 185 | 61,864 |
| M | PSOE | alejandrosolerm | 3412 | 798 | 208 | 1939 |
| M | PSOE | alfonsocendon | 2726 | 1470 | 812 | 7486 |
| M | PSOE | antidiofagundez | 254 | 4 | 4 | 0 |
| M | PSOE | apabellas | 163 | 11 | 8 | 0 |
| M | PSOE | arandapaco | 3521 | 1552 | 1218 | 3413 |
| M | PSOE | arnauramirez | 7134 | 514 | 324 | 6093 |
| M | PSOE | asanchog | 137 | 146 | 144 | 2 |
| M | PSOE | astro_duque | 522,984 | 159 | 41 | 29,125 |
| M | PSOE | cesarjramos | 9120 | 598 | 187 | 4246 |
| M | PSOE | conjosemfranco | 7058 | 1731 | 1662 | 5525 |
| M | PSOE | dioufluc | 1708 | 432 | 425 | 90 |
| M | PSOE | felipe_sicilia | 10,472 | 372 | 306 | 18,655 |
| M | PSOE | franciscopolo | 24,769 | 709 | 642 | 657 |
| M | PSOE | germanrenau | 1200 | 189 | 88 | 354 |
| M | PSOE | gomezdcelis | 9463 | 230 | 122 | 8309 |
| M | PSOE | guillermomeijon | 2795 | 765 | 597 | 1415 |
| M | PSOE | hectorgomezh | 5370 | 332 | 250 | 3720 |
| M | PSOE | javieranton | 1438 | 330 | 311 | 116 |
| M | PSOE | javiercerqueir4 | 252 | 317 | 125 | 578 |

**Table A1.** *Cont.*

| Gender | Party | Twitter Handle | Followers | Activity | Retweets | RT Times |
|--------|-------|----------------|-----------|----------|----------|----------|
| M | PSOE | javizqui | 5140 | 779 | 594 | 5678 |
| M | PSOE | jccampm | 6106 | 267 | 141 | 2381 |
| M | PSOE | jcduran_ | 5957 | 362 | 353 | 192 |
| M | PSOE | jfrserrano | 3536 | 902 | 541 | 1508 |
| M | PSOE | jlaceves | 2443 | 2889 | 2696 | 907 |
| M | PSOE | joseantoniojun | 405,094 | 573 | 417 | 45,650 |
| M | PSOE | josluisramosro2 | 235 | 131 | 130 | 0 |
| M | PSOE | jruizcarbonell | 10,373 | 262 | 235 | 187 |
| M | PSOE | juanb0462 | 386 | 146 | 137 | 9 |
| M | PSOE | juanluissotoadd | 2409 | 651 | 437 | 552 |
| M | PSOE | j_zaragoza_ | 48,567 | 418 | 11 | 158,209 |
| M | PSOE | lcsahuquillo | 1262 | 55 | 55 | 0 |
| M | PSOE | luisplanas | 13,176 | 328 | 159 | 6938 |
| M | PSOE | marclamua | 3097 | 280 | 228 | 591 |
| M | PSOE | migonzalezcaba | 1542 | 458 | 337 | 645 |
| M | PSOE | montimar66 | 1481 | 52 | 28 | 450 |
| M | PSOE | morissiero | 1273 | 1158 | 954 | 1513 |
| M | PSOE | nasholop | 5462 | 780 | 499 | 11,024 |
| M | PSOE | odonelorza2011 | 56,853 | 1118 | 192 | 28,813 |
| M | PSOE | pabloaranguena | 2428 | 519 | 51 | 15,358 |
| M | PSOE | patxilopez | 195,809 | 234 | 162 | 13,409 |
| M | PSOE | pedrosaurag | 5536 | 118 | 100 | 109 |
| M | PSOE | pedro_casares | 7061 | 996 | 538 | 17,958 |
| M | PSOE | perejoanpons | 4123 | 1181 | 634 | 1097 |
| M | PSOE | pmklose | 15,558 | 1837 | 913 | 13,581 |
| M | PSOE | salazarropaco | 5543 | 518 | 375 | 11,369 |
| M | PSOE | sanchezcastejon | 1,351,574 | 628 | 332 | 284,959 |
| M | PSOE | santicl | 4070 | 159 | 137 | 888 |
| M | PSOE | sarrimorell | 1853 | 231 | 222 | 3 |
| M | PSOE | sergio_gp | 7978 | 785 | 627 | 2055 |
| M | PSOE | simancasrafael | 25,695 | 654 | 500 | 44,649 |
| M | PSOE | valentingarciag | 4665 | 802 | 590 | 447 |
| M | PSOE | viondi | 6535 | 1890 | 571 | 103,622 |
| F | UP | ainavs | 14,655 | 516 | 322 | 7825 |
| F | UP | antonia_jover_ | 1260 | 156 | 49 | 216 |
| F | UP | gagupilar | 3569 | 254 | 102 | 3803 |
| F | UP | gloriaelizo | 20,056 | 742 | 498 | 21,838 |
| F | UP | ionebelarra | 69,252 | 204 | 141 | 20,034 |
| F | UP | isabel_franco_ | 13,447 | 407 | 204 | 14,110 |
| F | UP | lauralopezd | 2343 | 163 | 105 | 659 |
| F | UP | luciadalda | 2208 | 423 | 366 | 2616 |
| F | UP | margpuig | 5328 | 227 | 87 | 1458 |
| F | UP | maria_podemos | 1891 | 715 | 270 | 7168 |
| F | UP | marisasaavedram | 1623 | 501 | 279 | 2104 |
| F | UP | martinavelardeg | 4528 | 681 | 264 | 2563 |
| F | UP | roser_maestro | 3008 | 131 | 87 | 515 |
| F | UP | sofcastanon | 27,590 | 677 | 256 | 23,745 |
| F | UP | veranoelia | 48,083 | 142 | 93 | 8404 |
| F | UP | vickyrosell | 83,699 | 607 | 279 | 59,418 |
| F | UP | yolanda_diaz_ | 97,692 | 726 | 330 | 195,440 |
| M | UP | agarzon | 1,124,488 | 504 | 305 | 140,042 |
| M | UP | alber_canarias | 48,644 | 133 | 97 | 9804 |
| M | UP | antongomezreino | 14,567 | 1388 | 1067 | 21,902 |
| M | UP | ensanro | 29,365 | 474 | 219 | 51,465 |
| M | UP | eselkaos | 3459 | 1026 | 771 | 2969 |
| M | UP | g_pisarello | 41,505 | 816 | 542 | 27,690 |
| M | UP | hector_illueca_ | 7533 | 74 | 61 | 2398 |

**Table A1.** *Cont.*

| Gender | Party | Twitter Handle | Followers | Activity | Retweets | RT Times |
|--------|-------|----------------|-----------|----------|----------|----------|
| M | UP | ismael_cortesg | 1935 | 315 | 296 | 627 |
| M | UP | jaumeasens | 77,359 | 733 | 386 | 43,254 |
| M | UP | joanmena | 28,971 | 489 | 293 | 12,949 |
| M | UP | juralde | 83,061 | 1780 | 895 | 106,468 |
| M | UP | j_sanchez_serna | 11,223 | 363 | 253 | 27,173 |
| M | UP | mayoralrafa | 97,475 | 180 | 134 | 21,231 |
| M | UP | pnique | 536,961 | 1147 | 589 | 653,245 |
| M | UP | roberuriarte | 5169 | 142 | 20 | 2406 |
| M | UP | txemaguijarro | 4577 | 237 | 179 | 1335 |
| F | VOX | crisestebanvox | 5943 | 1767 | 759 | 13,653 |
| F | VOX | eledhmel | 62,742 | 419 | 66 | 137,998 |
| F | VOX | georgina_vox | 3057 | 534 | 429 | 6000 |
| F | VOX | lourdesmndezm1 | 12,489 | 564 | 520 | 10,974 |
| F | VOX | macarena_olona | 121,559 | 3176 | 2275 | 820,263 |
| F | VOX | malenanevado | 3473 | 563 | 250 | 6132 |
| F | VOX | meerrocio | 9977 | 1768 | 1267 | 71,004 |
| F | VOX | mestremanuel | 14,029 | 1865 | 1668 | 16,530 |
| F | VOX | patriciadlheras | 4050 | 448 | 195 | 9055 |
| F | VOX | rromerovilches | 15,720 | 980 | 708 | 17,268 |
| F | VOX | ruizsolas | 5416 | 39 | 13 | 1572 |
| F | VOX | teresagdvinuesa | 2839 | 953 | 890 | 5884 |
| F | VOX | _patricia_rueda | 8854 | 509 | 447 | 4883 |
| M | VOX | agustinrosety | 44,040 | 1612 | 659 | 234,249 |
| M | VOX | a_lopezmaraver | 1263 | 123 | 120 | 794 |
| M | VOX | cfdezrocysua | 5088 | 2016 | 1663 | 36,585 |
| M | VOX | czambranogr | 257 | 135 | 99 | 412 |
| M | VOX | edelvallerod | 4011 | 7767 | 6976 | 26,839 |
| M | VOX | fjconpe | 10,417 | 1064 | 545 | 19,733 |
| M | VOX | fjosealcaraz | 38,100 | 4383 | 2631 | 203,771 |
| M | VOX | igarrigavaz | 63,900 | 1462 | 962 | 120,222 |
| M | VOX | ivanedlm | 225,450 | 1422 | 962 | 544,576 |
| M | VOX | jlsteeg_doc | 417 | 11 | 9 | 0 |
| M | VOX | joaquinrobles55 | 1877 | 1763 | 1427 | 5934 |
| M | VOX | joseramirezdel2 | 8695 | 5260 | 4100 | 34,231 |
| M | VOX | juanjoaizcorbe | 4134 | 347 | 239 | 11,043 |
| M | VOX | luisgestoso | 4094 | 3386 | 2018 | 53,436 |
| M | VOX | mariscalzabala | 23,501 | 617 | 529 | 25,344 |
| M | VOX | mazureque | 235 | 2 | 2 | 0 |
| M | VOX | ortega_smith | 141,367 | 1021 | 923 | 100,778 |
| M | VOX | pablosaezam | 15,741 | 413 | 165 | 15,274 |
| M | VOX | pcalvoliste | 2101 | 1506 | 1306 | 5002 |
| M | VOX | pedro_fhz | 26,928 | 976 | 939 | 11,591 |
| M | VOX | rafalomana | 15,384 | 64 | 28 | 1564 |
| M | VOX | rchamode | 6781 | 3713 | 3017 | 11,707 |
| M | VOX | rodrijr111 | 2032 | 715 | 400 | 3917 |
| M | VOX | rubenmansolivar | 5357 | 683 | 68 | 9484 |
| M | VOX | sanchezdelreal | 50,262 | 4117 | 1848 | 255,674 |
| M | VOX | santi_abascal | 450,989 | 1048 | 752 | 1,427,478 |
| M | VOX | vicpiedra | 8739 | 561 | 393 | 11,788 |

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
