# Peer review of "Bridging the Gap: How Gender Influences Spanish Politicians’ Activity on Twitter"

_journalmedia, doi:10.3390/journalmedia2030028_

Round 1
Reviewer 1 Report
It is a study that addresses an interesting issue. Even so, the article could benefit from some improvements, as follows:
The article includes studies on the role of gender on Twitter, but lacks even a reference to those studies on the role of women politicians on Twitter.
- American Politics Research, 2016
-
Talking politics on Twitter: Gender, elections, and social networks
SC McGregor, RR Mourão - Social media+ society, 2016 - I am woman, hear me tweet! Gender differences in Twitter use among congressional candidates. KM Wagner, J Gainous, MR Holman - Journal of Women, Politics & …, 2017
- Etc.
The results of the QR are of interest, but they are not sufficiently argued or discussed in accordance with the contributions made to date in the literature on the subject.
What do these differences in the number of tweets, audience, amplification, etc., imply? What do the gender differences detected in twitter participation imply in the light of feminist theory and political theory, or even political communication theory?
With regard to the selection of the sample, is the selection of the dates of the analysis coincidental?
The results are abundant and interesting.
The findings state: "Our research has analysed the extent to which male and female members of the Spanish Parliament are equally influential on Twitter". However, this objective was not included in the RQs.
In order to increase the contribution of the research, it would have been interesting to analyse the content of the tweets, in order to delve deeper into the implications of these gender differences.
Probably the title should be adapted to fit the study carried out. Instead of How gener influences Spanish politics on Twitter, it should read How gender influences Spanish politicians' activity on Twitter.
Author Response
Dear reviewers and editors,
Thank you very much for addressing to us such valuable comments that, for sure, contribute to make our work better. Thank you for your time and suggestions.
We much appreciate all the bibliographical suggestions, most of them have been included, as the new version indicates. Please, bear in mind that McGregor and Mourao’s work have been worked and referred in the original version submitted, but we have considered your suggestion and add some more quotes.
Some of the reviewers indicate us what follows: “In order to increase the contribution of the research, it would have been interesting to analyse the content of the tweets, in order to delve deeper into the implications of these gender differences.” This is a very interesting point that we have been considering as a second part of the present paper. So, we think in keeping working on it including discourse analysis of the tweets.
We much appreciate the suggestion of modifying the title, that we’ve followed.
We’ve modified some aspects in the results as well as we have added the explanation of the selection of the sample.
Also, we’ve modified the point where the reviewer indicates: “The findings state: "Our research has analysed the extent to which male and female members of the Spanish Parliament are equally influential on Twitter". However, this objective was not included in the RQs.” Now we consider that the explanation is clear and we focus on the idea that influence is in terms of activity, amplification, audience and effectivity.
Thank you very much for your feedback, hope all these changes lead to the final approval.
Kind regards,
Authors
Reviewer 2 Report
Thank you for giving me the chance to review “Bridging the Gap. How Gender Influences Spanish Politics on Twitter”. The paper deals with an important and timely topic, thus, per se it represents an important contribution to existing literature.
The methodology is sound and results are very interesting.
However, this work presents one major flaw that should be solved prior to publication: the theoretical background is somehow confused and should be thoughtfully revised.
1) In order to systematize the theoretical framework, thoughtfully set the basis for the research questions, and make the reading easier it would be important to distinguish between general literature (international) and the specific Spanish case.
Thus, both when the Autho(s) speak about SSNN and political communication and women on SSNN it could be useful to start talking about international findings and then move to the specificities of the Spanish case.
2) SECTION 1.2. Communicating for Influence and Visibility
Section 1.2 should be reformulated. It should analyze in a deeper way the broad literature about political communication and social network, specifically Twitter.
Again, I suggest to start with international literature and then moving to the Spanish case.
In particular two concepts could be useful: 1) Americanization 2) Personalization of political communication
A reference that might be helpful:
CERVI, Laura; ROCA, Núria. Towards an Americanization of political campaigns? The use of Facebook and Twitter for campaigning in Spain, USA and Norway. Anàlisi, [S.l.], p. 87-100, jun. 2017. ISSN 2340-5236. doi: https://doi.org/10.5565/rev/analisi.3072.
In particular, this section does not take into consideration the wide selections of studies referring to Spain, that must be taken into consideration. There are myriad of studies. Andreu Casero, editor of this Journal, for instance, has published a lot of works on Twitter and political communication.
Other examples:
Cervi, L. y Roca Trenchs, N. (2018). El uso de Twitter por parte de los principales candidatos en las campañas electorales para las elecciones generales españolas: 2011 y 2015. ¿Brecha digital y generacional? Doxa Comunicación, 26, 99-126.
CERVI, Laura; ROCA, Nuria, La modernización de la campaña electoral para las elecciones generales de España en 2015. ¿Hacia la americanización? Comunicación y Hombre, núm. 13, enero, 2017, pp. 133-150
By starting with a broader discussion on how politician use Twitter, on the one hand the article would improve its general quality, on the other this would allow you to state your research question and justify it.
Minor issues:
- Whhy calling the introduction “Narrative Flow? Wouldn’t it be better “Introduction”?
- “Gender Differences in Political Power and Influence, and Underrepresentation and Empowerment in Politics”
And…and do not sound good: I suggest to rephrase it.
In addition, there are some minor linguistic flaws that an English editor will help solving.
Sure that the Author(s) will use these suggestions to improve the quality of their work I wish them good luck!
Author Response

(The authors gave the same response as above.)

Round 2
Reviewer 1 Report
I consider the changes made in the work to be interesting, but the research does not yet justify the interest of:
1) (RQ!1) To reveal differences in the use of Twitter between men and women in the political sphere and in Spain specifically (has this been done in other countries? In which ones? If it has not been analysed, what could be the reason? And if it has been analysed, what conclusions have been reached?
2) Publicise differences between parties. This may be interesting, but what does it have to do with the main objective of the study? This relationship is not clearly explained, this RQ2 is simply provided, and between the two there is a brief explanation that tries to clarify it, but it is not enough
3) Why is RQ1 not related to RQ3? If it is shown that this amplification is balanced, what does it implies or which would be the contribution of that result? It is totally descriptive, it is data that does not go beyond the mere evidence of a number and the authors should find a way to make these results and their research transcend and contribute. Describing phenomena is usually unoriginal and less valuable in our field.
Author Response
(RQ!1) To reveal differences in the use of Twitter between men and women in the political sphere and in Spain specifically (has this been done in other countries? In which ones? If it has not been analysed, what could be the reason? And if it has been analysed, what conclusions have been reached?
Yes, of course. This type of study has been done in other cases and settings that have not been in the Spanish case. Some of these examples are as follows, but our study is concentrated in media behavior which allows a picture of the main trends in the male and female interactions and productivity in Twitter. It has to be said that our study is based on the Spanish case, which allows a particular view that could be compare with different researches.
Yarchi, M. & Samuel-Azran, T. (2018) Women politicians are more engaging: male versus female politicians’ ability to generate users’ engagement on social media during an election campaign, Information. Communication & Society, 21:7, 978-995. This is a very interesting study which focuses on the tweets generated by male and female politicians during the 2015 Israeli campaign. This study allows us to know that female politicians posts a great number of posts in comparison to male politicians. In the Israeli case, social media provides a greater opportunity for female politicians to promote themselves. This is something that does not happen in the Spanish case.
Baeza Reyes, A. & Lamadrid Álvarez, S. (2015) Representations of women parliamentary candidates in new media. Cuadernos INFO, 39, 67-86. This study reveals a low female political representation in new media (during the 2013 electoral campaign in Chile), as well as the main topics made by women: gender issues or their reduction to family topics.
Beltran, J.; Gallego, A.; Huidobro, A.; Romero, E. & Padró, Ll. (2021) Male and female politicians on Twitter: a machine learning approach. European Journal of Political Research, 60, 239-251. Where the authors conclude the main topics related to male and female politicians. In the case of female, the main topics are about gender and social affairs, which indicates that the discourse in social media is stereotyped.
Demiarhan, K. & Çakir-Demirhan, D. (2015) Gender and Politics: Patriarchal discourse on social media. Public Relations Review, 41, 308-310. Where the authors support the idea that Twitter perpetuates the dominant discourses on society, and theses discourses is based on patriarchal ideology.
Publicise differences between parties. This may be interesting, but what does it have to do with the main objective of the study? This relationship is not clearly explained, this RQ2 is simply provided, and between the two there is a brief explanation that tries to clarify it, but it is not enough
As it is said throughout the manuscript, Party membership is also a predictor of politicians’ activity (Johnstonbaugh, 2020). Results of this research, but also previous research already cited in the manuscript (Verge and Troupel 2011; Verge & de la Fuente 2014), With RQ2 we wanted to measure the differences between parties, and between left- and right-wing parties, but considering the gender variable, as seen in Table 4 (Mean amount, amplification, efficacy and audience of Spanish Members by gender and party). In this sense, we consider that the question and the results are linked to the main objectives of the research.
Why is RQ1 not related to RQ3? If it is shown that this amplification is balanced, what does it implies or which would be the contribution of that result? It is totally descriptive, it is data that does not go beyond the mere evidence of a number and the authors should find a way to make these results and their research transcend and contribute. Describing phenomena is usually unoriginal and less valuable in our field.
Thank you for your appreciation, however we consider that the conclusions we reached are of interest especially because it conveys a certain way of behavior of the parties, with respect to Twitter, and also indicates how parties declared feminist respond, statistically, to a behavior parity. Undoubtedly, this work is a first step to a second stage where the analyze of the tweets discourse would indicate how the tend we have identify is communicated and spread in social media.
Reviewer 2 Report
The article has improved and it is now publishable.
Author Response
Thank you for your time and suggestions.